# Raphani Semen (*Raphanus sativus* L.) Ameliorates Alcoholic Fatty Liver Disease by Regulating De Novo Lipogenesis

**DOI:** 10.3390/nu13124448

**Published:** 2021-12-13

**Authors:** Woo Yong Park, Gahee Song, Joon Hak Noh, Taegon Kim, Jae Jin Kim, Seokbeom Hong, Jinbong Park, Jae-Young Um

**Affiliations:** 1Department of Science in Korean Medicine, Graduate School, Kyung Hee University, 26 Kyungheedae-ro, Dongdaemun-gu, Seoul 02447, Korea; parkwy0429@naver.com (W.Y.P.); gahee.ss1@khu.ac.kr (G.S.); 2Department of Pharmacology, College of Korean Medicine, Kyung Hee University, 26 Kyungheedae-ro, Dongdaemun-gu, Seoul 02447, Korea; thejinbong@khu.ac.kr; 3College of Korean Medicine, Kyung Hee University, 26 Kyungheedae-ro, Dongdaemun-gu, Seoul 02447, Korea; silverayne8@gmail.com (J.H.N.); iighst14@hanmail.net (T.K.); qkfkaehtlfkr@naver.com (J.J.K.); 4Department of Korean Medicine, Graduate School, Kyung Hee University, 26 Kyungheedae-ro, Dongdaemun-gu, Seoul 02447, Korea; hsb0513@naver.com

**Keywords:** AFLD, Raphani Semen, lipogenesis, SREBF1, Lpin1

## Abstract

In this study, we investigated the pharmacological effect of a water extract of Raphani Semen (RSWE) on alcoholic fatty liver disease (AFLD) using ethanol-induced AFLD mice (the NIAAA model) and palmitic acid (PA)-induced steatosis HepG2 cells. An RSWE supplement improved serum and hepatic triglyceride (TG) levels of AFLD mice, as well as their liver histological structure. To explore the molecular action of RSWE in the improvement of AFLD, we investigated the effect of RSWE on four major pathways for lipid homeostasis in the liver: free fatty acid transport, lipogenesis, lipolysis, and β-oxidation. Importantly, RSWE decreased the mRNA expression of de novo lipogenesis-related genes, such as Srebf1, Cebpa, Pparg, and Lpin1, as well as the protein levels of these factors, in the liver of AFLD mice. That these actions of RSWE affect lipogenesis was confirmed using PA-induced steatosis HepG2 cells. Overall, our findings suggest that RSWE has the potential for improvement of AFLD by inhibiting de novo lipogenesis.

## 1. Introduction

Alcoholic fatty liver disease (AFLD), characterized by excessive fortification of triglycerides in the liver, is one of the major causes of the wide spectrum of hepatic pathologies, such as hepatitis, steatohepatitis, fibrosis cirrhosis, and hepatocellular carcinoma (HCC) [1]. The dysfunction of alcohol metabolism in the liver caused by chronic heavy drinking is known to be a major cause of AFLD. However, because only about 20% of patients with AFLD are verified to have high alcohol dependence, the precise mechanisms leading to AFLD are still unclear [2]. Alcohol, but also age, gender, genetic background, nutritional status, occupational hazards, and viral diseases especially Hepatitis C Virus (HCV) infections, have been suggested as potential risk factors of AFLD [3]. The effective treatment strategies proposed for AFLD should be accompanied by additional treatment strategies, as well as restrictions on alcohol consumption.

Lipid homeostasis in the liver is maintained by regulating de novo lipogenesis, oxidation, and transport of fatty acids. In AFLD, ethanol-derived metabolites cause damage ranging from lipid accumulation in hepatocytes to inflammation, fibrosis, and carcinogenesis [4]. When a body is alcohol-fed, triglyceride (TG) levels are elevated, and more free fatty acids (FFA) are released into the circulatory system [5]. Excess circulating FFA can be delivered to the liver. Fatty acids are either oxidized by mitochondrial β-oxidation or incorporated into TG, resulting in lipid accumulation in the liver [6]. Many factors mediate lipid homeostasis in the liver by regulating de novo lipogenesis, oxidation, and transport of fatty acids. Abundant evidence suggests that hepatic lipid metabolism is modulated by critical regulators, such as sterol regulatory element binding proteins (SREBPs) and AMP-activated protein kinase (AMPK) [7,8]. Excessive lipid accumulation in hepatocytes results in massive death of these hepatocytes. The massive die-off triggers pro-inflammatory and pro-fibrogenic responses that increase the risks of liver cancer [9]. Given that hepatocellular lipid accumulation is the earliest sign of alcoholic liver disease, further study on the regulation of lipid metabolism in hepatocytes could create opportunities for early therapeutic intervention for those at risk of advanced disease [4].

Raphani Semen, the seeds of *Raphanus sativus* L., has been used as a medicinal herb to improve gastrointestinal disorders, such as indigestion, inflammation, and diarrhea in Korean Medicine [10]. Raphani Semen contains an extensive variety of pharmaceutically active compounds [11]. In fact, advanced information about the pharmacological role of Raphani Semen in relation to several gastrointestinal diseases has been revealed in a variety of animal experiment models, e.g., ulcerative colitis, intestinal motility, and colon cancer [12,13]. In addition, Raphani Semen and its active compounds have been shown to have potential for antioxidant and anti-inflammatory activity [14,15]. Here, we report on our investigation of whether Raphani Semen has a therapeutic effect on AFLD using the National Institute on Alcohol Abuse and Alcoholism (NIAAA) mouse model and PA-induced steatosis HepG2 cells, with a focus on the regulation of lipid metabolism.

## 2. Materials and Methods

### 2.1. Drugs and Reagents

Dulbecco’s modified Eagle’s medium (DMEM) with low glucose, penicillin–streptomycin, and fetal bovine serum (FBS) were purchased from Gibco BRL (Grand Island, NY, USA). Palmitic acid was purchased from Sigma-Aldrich (St. Louis, MO, USA). The Lieber-DeCarli alcohol liquid diet was obtained from Bio-Serv (Frenchtown, NJ, USA). The antibodies of C/EBPα, SREBP1, Lipin-1, p-IκBα, and NF-κB were purchased from Santa Cruz Biotechnology (Dallas, TX, USA). GAPDH, β-actin, p-ACC, and PPARγ antibodies were purchased from Cell Signaling Technology (Danvers, MA, USA). The antibodies of HSL, ATGL, and CPT1β were obtained from Abcam plc. (Hills Road, Cambridge, UK).

### 2.2. Sample Preparation

Dried Raphani Semen was purchased at the Han Yak Jae Market (Seoul, Korea). Extraction of Raphani Semen was performed based on its traditional use as a medicinal herb. The water extract of Raphani Semen (RSWE) was obtained by extracting Raphani Semen in hot water at 100 °C for 3 h, followed by filtering (Whatman, Kent, UK). After being freeze-dried in a vacuum, the solid was dissolved in distilled water (20 mg/mL). The extraction yield was 12.8% (*g*/*g*).

### 2.3. Ethical Statement

All animal experiments were performed in accordance with the ethical guidelines of Kyung Hee University and approved by the Institutional Review Board of Kyung Hee University (confirmation number: KHUASP (SE)-15-08).

### 2.4. Animal Experiments

Here, to induce the AFLD mice model, we chose the NIAAA model. The NIAAA model is a simple mouse model of alcoholic liver injury induced by chronic ethanol feeding (10 d ad libitum oral feeding with the ELD) plus single-binge ethanol feeding [16]. The histological condition produced by ELD and 35% EtOH-feeding was observed in the AFLD mice model.

Four-week-old male C57BL/6J mice were purchased from the Dae-Han Experimental Animal Center (Dae-Han Biolink, Eumsung, Korea) and housed under a 12 h light/dark cycle at a humidity of 70% and a constant temperature of 23 ± 2 °C. Voluntary liquid diet (LD) and tube feeding through the control Lieber–DeCarli diet were applied for the first five days. Afterward, for 10 days, the ethanol-fed groups (*n* = 4), the ethanol-fed and RSWE (100 mg/kg/day)-fed group (*n* = 4) voluntarily ingested a 5% ethanol Lieber–DeCarli (ELD) diet (Bio-Serv, NJ, USA), while the normal control group (*n* = 4) was fed a diet with identical calories. After maintaining the ethanol liquid diet to induce a chronic AFLD model, 30% EtOH was orally administered 9 h prior to sacrifice on the 11th day to induce acute liver toxicity. Mice fed a standard laboratory diet (CJ Feed Co., Ltd., Seoul, Korea) for 14 d were used as the normal control group. The animals were given free access to food and tap water. The body and food intake were recorded every week. At the end of this period, the animals were anesthetized under 30% CO_2_ asphyxiation, and serum was separated immediately after blood collection. Tissue samples were collected, placed in a tube, and stored at −80 °C.

### 2.5. Blood Serum Analysis

Serum TG, alanine transaminase (ALT), aspartate aminotransferase (AST), blood urea nitrogen (BUN), and creatinine levels were analyzed using enzymatic colorimetric methods at the Seoul Medical Science Institute (Seoul Clinical Laboratories, Seoul, Korea).

### 2.6. Cytokine Measurement

Hepatic cytokine levels were measured using Mouse IL-6 or TNF-α ELISA kit (Invitrogen, Waltham, MA, USA). The levels of IL-6 or TNF-α were confirmed based on the manufacturer’s instructions.

### 2.7. Hepatic TG Measurement

Triglyceride contents were detected using the EZ-Triglyceride Quantification kit (DoGen Bio, Seoul, Korea). These levels were confirmed based on the manufacturer’s instructions.

### 2.8. Hepatic FFA Measurement

Free fatty acid contents were detected using the EZ-Free Fatty Acid kit (DoGen Bio, Seoul, Korea). These levels were confirmed based on the manufacturer’s instructions.

### 2.9. Hematoxylin and Eosin (H&E) Staining

H&E staining was performed as previously reported [17]. Briefly, the liver tissues were fixed in 10% formalin and embedded in paraffin. According to a standard protocol, tissues were cut into 4 μm sections. The sections were then stained with H&E. Microscopic examinations were performed and photographs were taken under a regular light microscope.

### 2.10. Protein Extraction and Western Blot Analysis

Protein extraction and western blot analysis were performed as previously reported [18]. Briefly, protein extracts from homogenized liver or harvested HepG2 cells were lysed in radioimmunoprecipitation assay (RIPA) buffer (Cell Signaling Technology, Danvers, MA, USA) and the protein concentration was determined. The lysates were resolved by sodium dodecyl sulfate–polyacrylamide gel electrophoresis and transferred onto PVDF membranes. The membranes were blocked and incubated with primary antibodies (1:1000), followed by incubation with horseradish peroxidase-conjugated secondary antibodies (1:10,000). The protein signals were detected using an EZ-Western Lumi Femto Kit (DoGenbio, Seoul, Korea).

### 2.11. RNA Isolation and Quantitative PCR (qPCR)

Total RNA was extracted using a GeneAllR RiboEx total RNA extraction kit (GeneAll Biotechnology, Seoul, Korea). Newly synthesized complementary DNA (cDNA) from liver or HepG2 cells was amplified using specific primers and the Fast SYBR Green PCR master mix (Applied Biosystems, Foster City, CA, USA). mRNA expression was measured with a StepOnePlus qPCR System and StepOne Software v2.1 (Applied Biosystems, Foster City, CA, USA). The primers used in the experiments are shown in Table 1.

### 2.12. Immunofluorescence (IF) Assay

An IF assay was performed as previously reported [19]. For IF staining, liver tissues were fixed with 4% paraformaldehyde in PBS for 15 min and permeabilized with 0.2% Triton X-100 (Sigma-Aldrich) for 10 min. Thereafter, nonspecific binding sites were blocked using PBS with 1% bovine serum albumin (Calbiochem, San Diego, CA, USA). After incubation with primary antibodies for either SREBP1 (1:200) or Lipin-1 (1:500), the slides were incubated with Alexa Fluor 633 (1:500; Thermo Fisher Scientific) as secondary antibody for SREBP1, or with Alexa Fluor 488 (1:500; Thermo Fisher Scientific) as secondary antibody for Lipin-1. Fluorescence signals were measured by flow cytometry and imaged with the EVOSR Cell Imaging System (Thermo Scientific, Carlsbad, CA, USA).

### 2.13. Cell Culture and PA-Induced Lipid Accumulation in HepG2 Cells

HepG2 cell culture was performed as previously described [18]. The cells were cultured at 37 °C under 5% CO_2_ in DMEM with low glucose containing 10% FBS and 1% penicillin–streptomycin. To induce excessive lipid accumulation, HepG2 cells were seeded at 5 × 10^5^ cells/well in a 6-well culture plate and incubated for 24 h. On day 2, they were pretreated with RSWE (250 or 500 μg/mL) for 30 min, and then 150 μM PA was added to each well for 24 h. To prepare the PA, sodium palmitate was conjugated with a culture medium containing 1% bovine serum albumin (BSA).

### 2.14. Cell Cytotoxicity

To assess the cytotoxicity of RSWE, HepG2 cells (2.5 × 10^3^ cells/200 μL) were seeded in 96-well microplates and treated with various concentrations of RSWE (0–1000 μg/mL). After 24 h, the cells were incubated with 20 μL of WST-1 solution (Biomax Co., Ltd., Seoul, Korea) for 4 h at 37 °C under 5% CO_2_ and 95% air. The absorbance was measured at 440 nm using a VERSAmax microplate reader (Molecular Devices, Sunnyvale, CA, USA).

### 2.15. LDH Measurement

To assess the protective effect on the PA-induced LDH release of RSWE, HepG2 cells (2 × 10^4^ cells/200 μL) were seeded in 96-well microplates and treated with various concentrations of RSWE (0 to 500 μg/mL) or PA (150 μM). After 24 h, the cells were incubated with 20 μL of LDH solution (DoGen Bio, Seoul, Korea) for 30 min at 37 °C under 5% CO_2_ and 95% air. The absorbance was measured at 450 nm using a VERSAmax microplate reader (Molecular Devices, Sunnyvale, CA, USA).

### 2.16. BODIPY^TM^ Staining

For immunofluorescent staining of BODIPY^TM^ (Thermo Scientific), HepG2 cells were treated with RSWE or PA and stained with 2 µM BODIPY^TM^ for 30 min at 37 °C, and fluorescence was detected using the EVOSR Cell Imaging System (Thermo Scientific). For flow cytometry analysis with BODIPY^TM^, BODIPY^TM^ (green) fluorescence was measured in the total event count (7 × 10^3^ cell/50 uL).

### 2.17. Oil Red O Staining

The accumulation of intracellular lipid was measured using Oil Red O [20], of which the working solution was prepared as described. The HepG2 cells were fixed with 10% formalin and then stained for 1 h with a filtered Oil Red O solution. To quantify the intracellular lipids, the stained lipid droplets were dissolved in 100% isopropanol (3 mL per well). The extracted dye was transferred to a 96-well plate, and the absorbance was read with a VERSAmax microplate reader (Molecular Devices, Sunnyvale, CA, USA) at 500 nm.

### 2.18. Statistical Analysis

Data are expressed as the mean ± standard error of the mean (S.E.M.). Significant differences between groups were determined using the Student’s *t*-test or one-way ANOVA followed by post hoc Tukey’s multiple comparison tests. Statistical differences were determined using a subsequent post hoc one-tailed Mann–Whitney U test and calculated using Prism 8 (GraphPad Software, San Diego, CA, USA). The statistical significance of differences was presented as one of the probability values: *p* < 0.05 and *p* < 0.01.

## 3. Results

### 3.1. RSWE Attenuates Fatty Liver Phenotypes in AFLD Mice

To explore the effect of RSWE on AFLD, we used the NIAAA model. In brief, the mice were randomly divided into three groups of four mice each, as follows: LD group (LD-fed), AFLD group (ELD and 35% EtOH-fed), and RSWE group (ELD and 35% EtOH-fed with RSWE 100 mg/kg, per oral). The detailed scheme is shown in Figure 1A. We first checked the effect of RSWE on the AFLD phenotypes of the AFLD mice. Body weights and liver weights of the AFLD group were significantly decreased compared to the LD group, but RSWE treatment had no effect on the reduced body and liver weights (Figure 1B). The levels of plasma parameters for liver or kidney injury were increased in the AFLD group and were significantly decreased by RSWE treatment. Meanwhile, the decreased BUN in the AFLD group was not restored by RSWE treatment (Figure 1C). The high levels of pro-inflammatory mediators (TNF-a, IL-6, p-IκBα, and NF-κB), major pathological makers of AFLD, were reduced by treatment with RSWE (Figure 1D and Appendix A).

As shown in Figure 1E, the hepatic tissues of AFLD mice displayed pathological changes, such as larger lipid droplets, suggesting that alcohol-induced liver injury was successfully established in the AFLD mice. RSWE treatment restored the histological structure of the liver. In addition, the increased hepatic TG, FFA, and serum TG contents in AFLD mice were decreased by RSWE treatment (Figure 1F,G). These findings suggest that RSWE improves the alcohol-induced fatty liver phenotypes of mice.

### 3.2. RSWE Regulates De Novo Lipogenesis in the Liver of AFLD Mice

The factors related to fatty acid transport (hepatic lipid uptake/export), de novo lipogenesis, and lipolysis/oxidation were confirmed in the hepatic tissues of AFLD mice using qPCR analysis. Most of the hepatic lipid metabolism-related genes changed under exposure to alcohol, but lipogenesis-related genes were only controlled by RSWE treatment. In detail, RSWE treatment reduced lipogenesis-related genes, such as *Srebf1*, *Fas*, *Cebpa*, *Pparg*, *Lpin1,* and *Acc*, but had no effect on fatty acid transport factors (*Cd36*), lipolysis factors (*Atgl* and *Hsl*), or β-oxidation factors (*Cpt1a*, *Ppara*, and *Cyp2e1*) (Figure 2A and Appendix A). Regulation by RSWE of lipogenic factors was also confirmed at the protein level in the hepatic tissue of AFLD mice. As shown in Figure 2B,C, RSWE treatment decreased the levels of lipogenesis-related proteins that included PPARγ, C/EBPα, SREBP1, and Lipin-1. This was confirmed through cytological analysis using immunofluorescence staining (Figure 2D,E). Interestingly, RSWE treatment enhanced the protein levels of HSL and CPT1β in the liver of AFLD mice (Appendix A). These results suggest that RSWE operates to reduce lipogenesis-associated factors during the development and progression of ALFD.

### 3.3. RSWE Inhibits Lipid Accumulation in PA-Induced Steatosis HepG2 Cells

We observed that RSWE attenuates lipogenesis in the liver of AFLD mice. To confirm the inhibitory effect of RSWE on lipogenesis in fatty liver, we used the PA-induced steatosis HepG2 cell model for in vitro studies. The cytotoxicity of RSWE was first measured in HepG2 cells. The cells were treated with RSWE (125–1000 μg/mL), and then the WST-1 assay was performed. RSWE showed cytotoxicity at 1000 μg/mL of concentration in HepG2 cells (Figure 3A). Cytotoxicity of RSWE treated with PA was also confirmed. When the cells were treated with RSWE (125–1000 μg/mL) in PA (150 μM)-treated HepG2 cells, treatment with 125, 250, and 500 μg/mL of RSWE significantly increased cell viability (Figure 3B). Thus, we chose two concentrations (250 and 500 μg/mL) for further experiments. In addition, to investigate the inhibition effect on PA-induced LDH releases of RSWE, we treated HepG2 cells with PA or RWSE (125, 250 and 500 μg/mL). RSWE treatment significantly reduced LDH releases in the supernatant compared to PA-treated HepG2 cells (Figure 3C). Next, TG accumulation was measured using BODIPY^TM^ and Oil Red O staining. The intensity of BODIPY^TM^ was measured by a fluorescent microscope and flow cytometry, and RSWE treatment (250 and 500 μg/mL) was shown to reduce intracellular TG levels in PA-treated HepG2 cells (Figure 3D–F). In Oil Red O staining, RSWE treatment (500 μg/mL) significantly decreased lipid accumulation in PA-treated HepG2 cells (Figure 3G,H). Furthermore, qPCR results also showed that the expression of lipogenesis-associated genes, including *PPARG*, *CEBPA*, *SREBF1*, *APOB*, and *FABP4*, significantly decreased in RSWE (250 or 500 μg/mL)- and PA-treated HepG2 cells (Figure 3I). The protein levels of PPARγ and C/EBPα were also confirmed (Figure 3J).

### 3.4. RSWE Regulates SREBP1 and Lipin-1 in PA-Induced Steatosis HepG2 Cells

Just as for the in vivo results, RSWE also inhibited SREBP1 and Lipin-1, major factors contributing to the de novo lipogenesis pathway. From western blot and IF staining analyses, we confirmed that 250 and 500 μg/mL of RSWE decreased SREBP1 and Lipin-1 proteins in PA-treated HepG2 cells. In particular, 500 μg/mL of RSWE decreased the levels of SREBP1 and Lipin-1 proteins by 0.45 ± 0.25-fold and 0.33 ± 0.23-fold, respectively, compared to PA-treated HepG2 cells (Figure 4A,D). Likewise, the expression of SREBP1 and Lipin-1 decreased RSWE concentrations of 250 and 500 μg/mL (Figure 4B,C,E). Overall, these findings showed that RSWE can inhibit lipogenesis-related factors, including SREBP1 and Lipin-1, suggesting that RSWE has potential as a candidate agent for AFLD treatment.

## 4. Discussion

Alcohol exposure is one of the potential causes of a wide spectrum of hepatic pathologies, including alcoholic fatty liver disease [21]. The degree of damage to the alcoholic fatty liver varies depending on the amount of alcohol consumed and the frequency of drinking [22]. Alcoholic fatty liver disease may progress to hepatitis, and in severe cases cirrhosis can occur. The amount of alcohol and the frequency of alcohol consumption act as important factors in the degree of alcoholic liver damage [23]. Characterized by histological lesions, including liver steatosis, or even cirrhosis in severe cases, any form of AFLD can lead to end-stage liver diseases, according to long-term studies of biopsy specimens and patient outcomes [24]. The current standard treatment for alcoholic hepatitis is based on steroids [25]. Liver transplantation treatment may be performed for non-targeted liver cirrhosis patients who are willing to abstain from drinking. However, effective and safe therapeutic options are still limited for AFLD. In fact, the application of propylthiouracil, colchicine, antioxidants, and phosphatidylcholine have been reported as drug treatments for AFLD, but their therapeutic effects have not proven very successful [26]. Recently, natural products and isolated compounds with relatively few side-effects have been found to have beneficial effects on AFLD. This suggests that these may be considered to have potential for development as substitute therapeutic agents [27]. Therefore, the present study was conducted to evaluate the beneficial action of RSWE, which has been used as a medicinal herb for treatment of AFLD.

In the present study, RSWE reduced the number of lipid droplets and cracks in the livers of AFLD mice. In particular, RSWE decreased the representative phenotypes of alcohol-induced fatty liver disease, which include increases in hepatic and serum TG levels in AFLD mice. Elevated FA synthesis is a common phenomenon observed in chronic alcohol drinkers [28]. The liver releases some of these fatty acids into the blood in the form of very low-density lipoproteins (VLDLs) [29]. However, when this continues and when the rate of TG synthesis in the liver exceeds the rate of VLDL release, the liver starts to accumulate lipids [30]. Moreover, RSWE also reduced the serum levels of TG, ALT, AST, and creatinine. However, the BUN was not changed. ALT and AST are the most frequently used biomarkers to evaluate liver function. From these results, it appeared to us that RSWE had anti-AFLD activity in the AFLD mice model.

Alcoholic fatty liver disease develops through four major mechanisms. First, alcohol increases FA synthesis and lipogenesis [28]. Second, alcohol promotes the mobilization of FAs and lipids from adipose tissue and the intestine to the liver [31]. Third, alcohol inhibits FA β-oxidation via the inactivation of PPARα and downstream β-oxidation genes via the elevation of acetaldehyde, adenosine, and cytochrome P450 2E1 (CYP2E1) [32]. Lastly, alcohol inhibits CPT1 activity [33]. To confirm the way in which RSWE inhibits AFLD, we measured mRNA levels of lipid metabolism-associated factors in fatty livers, including lipogenesis, FA transport, β-oxidation, and lipolysis in the livers of AFLD mice. RSWE significantly regulates mRNA expression levels of lipogenesis-associated genes, such as *Srebf1, Pparg, Fas, ApoB,* and *Cpt1b*. RSWE also reduces the protein expression levels of SREBP1, Lipin-1, PPARγ, and C/EBPα. In particular, SREBP1 and Lipin-1 are important transcriptional regulators in de novo lipogenesis [34]. SREBP1 is transferred from the endoplasmic reticulum to the Golgi apparatus by insulin stimulation, processed, and then transferred to the nucleus to induce a group of genes involved in cholesterol and fatty acid synthesis [35]. Lipin-1 also has dual functions in lipid metabolism. In hepatocytes of AFLD, alcohol increases the expression of Lipin-1 through AMPK-SREBP1 signaling [36]. Lipin-1 acts as a co-activator of PPARα in the liver and is involved in fatty acid oxidation [37]. PPARγ and C/EBPα in the liver induce the adipogenic program to store fatty acids in lipid droplets, as observed in adipocytes [38]. Our results show that RSWE alleviates AFLD through the regulation of lipogenesis-related factors.

In conclusion, our findings suggest the novel finding that RSWE improves AFLD in PA-induced steatosis HepG2 cells and ethanol-induced AFLD mice through the inhibition of lipogenesis. However, our findings have limitations because information about the chemical profiling of RSWE and its efficacy for AFLD treatment is excluded. We only focused on the pharmacological effects of RSWE; further investigation is required to confirm the underlying mechanism and to find the active compounds of RSWE. Nevertheless, the outcome of the study presented here indicates an effective therapeutic approach for the treatment of fatty liver disease. We also propose that a better understanding of RSWE could open up possibilities for regulating lipogenesis in patients with AFLD and AFLD-related metabolic diseases using new drugs or even dietary supplementation.

## Figures and Tables

**Figure 1 nutrients-13-04448-f001:**
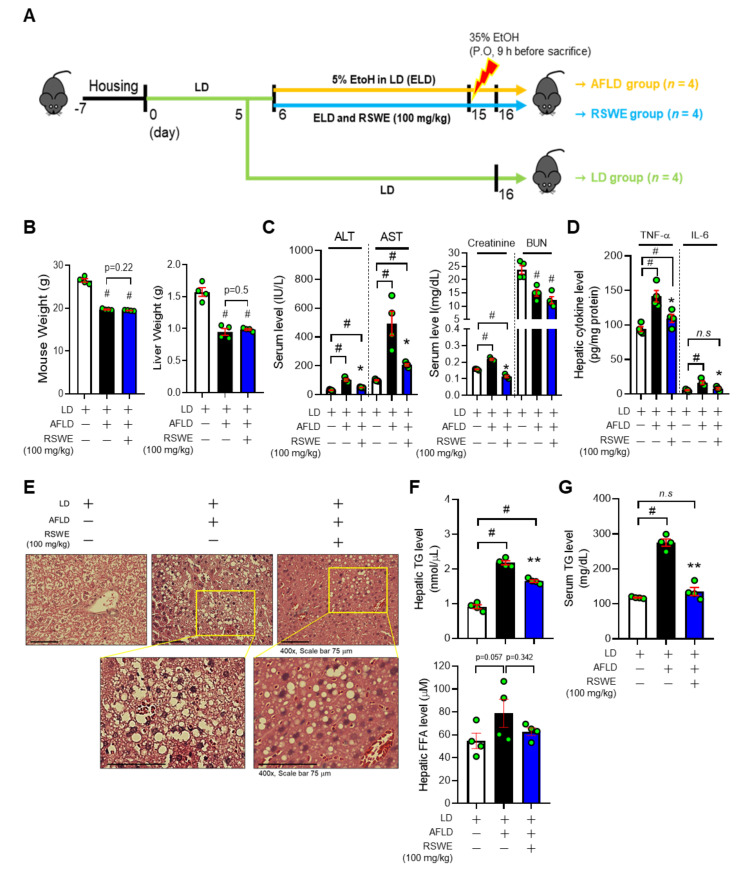
The effect of RSWE on AFLD phenotypes in the C57BL/6J NIAAA mouse model. (**A**) The experimental procedure. (**B**) Body weight and liver weight were measured. (**C**) Levels of ALT, AST, creatinine, and BUN in serum were measured. (**D**) Levels of TNF-α and IL-6 in hepatic tissue were measured using ELISA kits. (**E**) Paraffin-embedded liver was stained with H&E (magnification ×400, scale bar 75 μm). (**F**) TG and FFA levels of hepatic tissue were measured. (**G**) TG levels of serum were measured. All data are expressed as the mean ± S.E.M. of data from three or more separate experiments. Statistical differences were determined using a subsequent post hoc one-tailed Mann–Whitney U test. *n.s.* no significance, # *p* < 0.05 vs. LD group and * *p* < 0.05, ** *p* < 0.01 vs. AFLD group. RSWE: Raphani Semen water extract, LD: liquid diet, AFLD: alcoholic fatty liver disease, ALT: alanine transaminase, AST: aspartate transaminase, BUN: blood urea nitrogen, TG: triglycerides, FFA: free fatty acid.

**Figure 2 nutrients-13-04448-f002:**
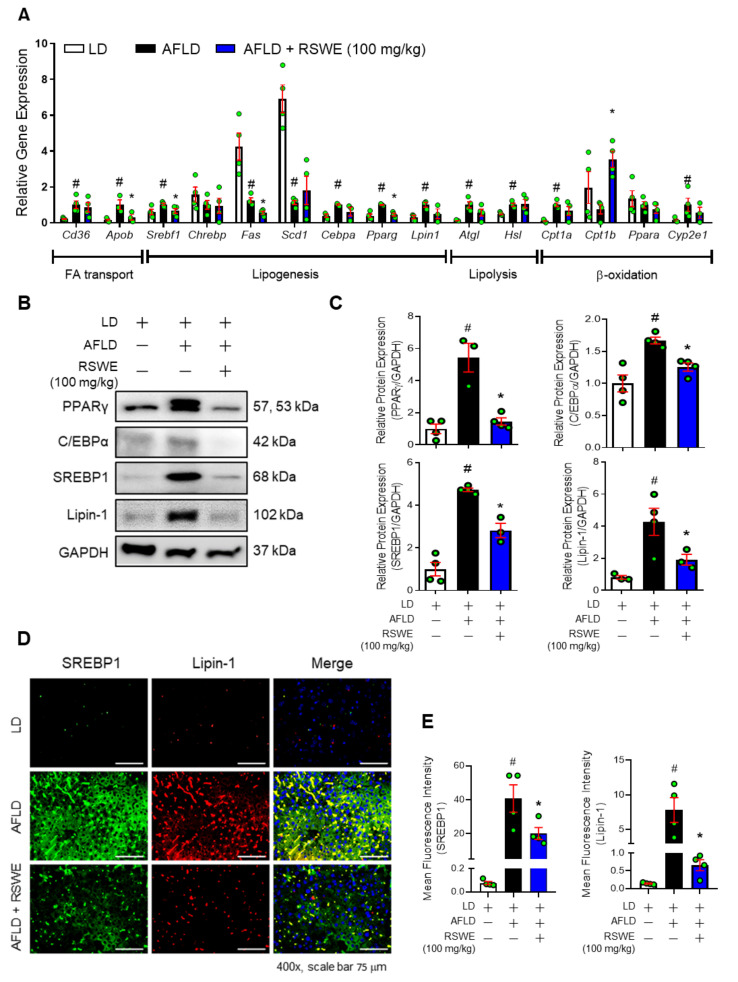
The inhibitory effect of RSWE on lipogenesis in the liver of a C57BL/6J NIAAA mouse model. (**A**) mRNA expression of lipid metabolism-related genes in the liver was analyzed using qPCR. (**B**,**C**) Protein levels of PPARγ, C/EBPα, SREBP1, and Lipin-1 were analyzed using western blot analysis. GAPDH was used as a loading control. (**D**,**E**) SREBP1 (green), Lipin-1 (red), and nuclei (blue) were detected in hepatic tissue using IF staining (magnification ×400, scale bar 75 μm). Fluorescence intensity was quantified using ImageJ software. All data are expressed as the mean ± S.E.M. of data from three or more separate experiments. Statistical differences were determined using a subsequent post hoc one-tailed Mann–Whitney U test. # *p* < 0.05 vs. LD group and * *p* < 0.05 vs. AFLD group. RSWE: Raphani Semen water extract, LD: liquid diet, AFLD: alcoholic fatty liver disease.

**Figure 3 nutrients-13-04448-f003:**
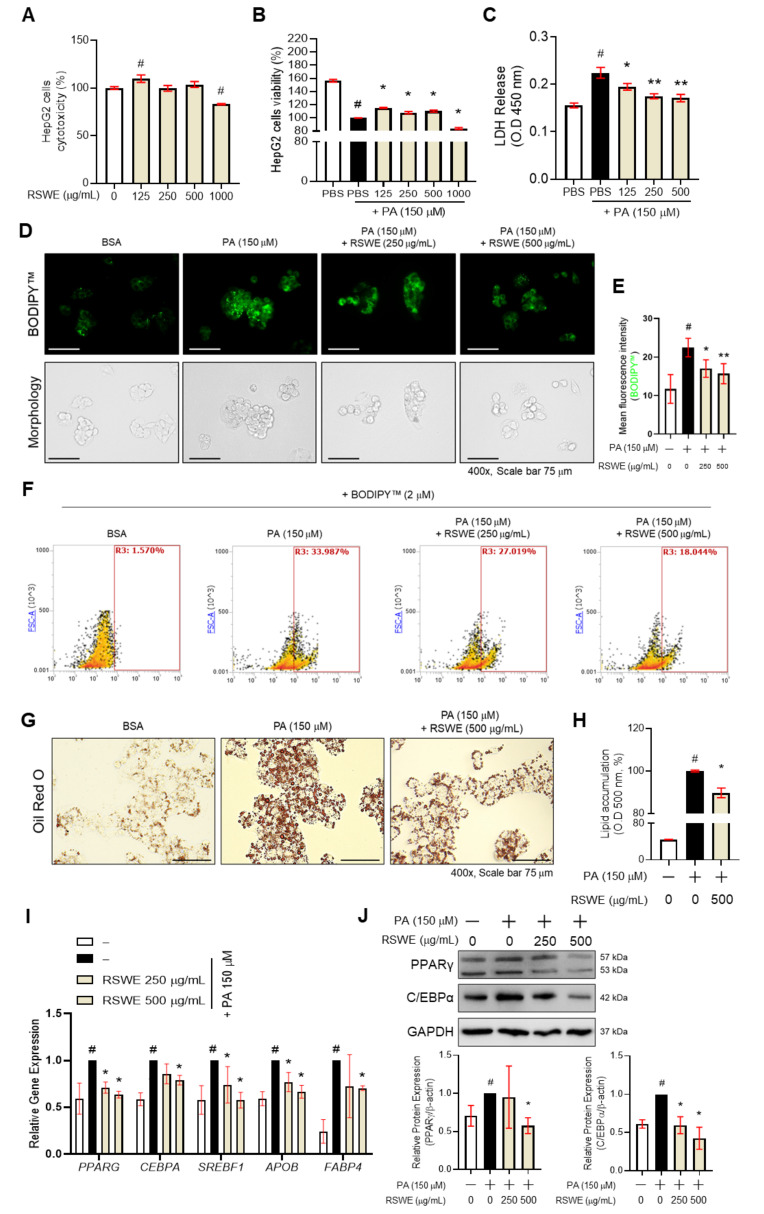
The effect of RSWE on lipid accumulation in PA-treated HepG2 cells. (**A**) Cytotoxicity and (**B**) cell viability of RSWE with or without PA on HepG2 hepatocytes was measured with a WST-1 assay. (**C**) LDH release was measured using an LDH assay kit. (**D**–**F**) Intracellular TG levels were analyzed using BODIPY^TM^ staining. (**G**,**H**) Lipid accumulation was measured using Oil Red O staining. (**I**) mRNA expression of lipogenesis-related genes, including *PPARG, CEBPA, SREBF1*, *APOB*, and *FABP4*, in HepG2 hepatocytes was analyzed using qPCR. (**J**) Protein levels of PPARγ and C/EBPα were analyzed using western blot analysis. GAPDH was used as a loading control. All data are expressed as the mean ± S.E.M. of data from three or more separate experiments. Statistical differences were determined using a subsequent post hoc one-tailed Mann–Whitney U test. # *p* < 0.05 vs. PBS-treated HepG2 cells and * *p* < 0.05 and ** *p* < 0.01 vs. PA-treated HepG2 cells. RSWE: Raphani Semen water extract, PA: palmitic acid.

**Figure 4 nutrients-13-04448-f004:**
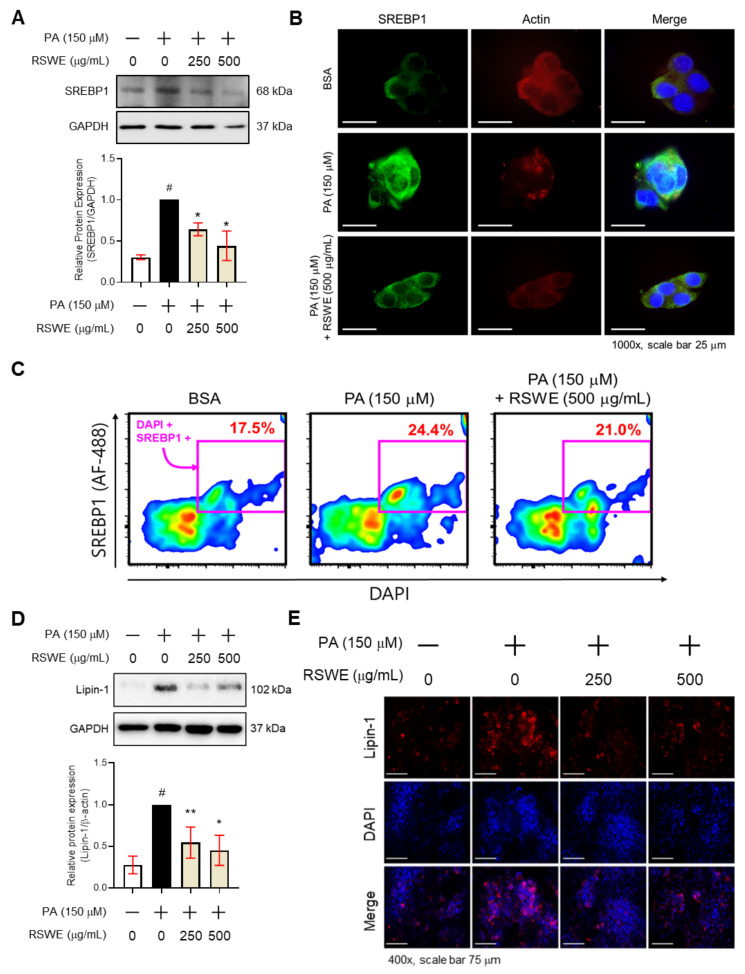
The regulatory effect of RSWE on SREBP1 and Lipin-1 expression in PA-treated HepG2 cells. (**A**) SREBP1 protein was analyzed using western blot analysis. (**B**) SREBP1 (green), actin (red) and nuclei (blue) were detected in HepG2 hepatocytes using IF staining (magnification ×400, scale bar 75 μm). (**C**) The rates of co-localization of SREBP1 (green) and DAPI (Blue) were calculated by flow cytometry. (**D**) Lipin-1 protein was analyzed by western blot analysis. GAPDH was used as a loading control. (**E**) Lipin-1 (red) and nuclei (blue) were detected in HepG2 hepatocytes using IF staining (magnification ×400, scale bar 75 μm). All data are expressed as the mean ± S.E.M. of data from three or more separate experiments. Statistical differences were determined using a subsequent post hoc one-tailed Mann–Whitney U test. # *p* < 0.05 vs. PA-treated HepG2 cells and * *p* < 0.05 and ** *p* < 0.01 vs. PA-treated HepG2 cells. RSWE: Raphani Semen water extract, PA: palmitic acid.

**Table 1 nutrients-13-04448-t001:** Primer sequences used for qPCR.

Genes	Forward (5′ to 3′)	Reverse (5′ to 3′)
*m-Apob*	TTGGCAAACTGCATAGCATCC	TCAAATTGGGACTCTCCTTTAGC
*m-Atgl*	ATATCCCACTTTAGCTCCAGGG	CAAGTTGTCTGAAATGCCGC
*m-Cd36*	ATGGGCTGTGATCGGAACTG	GTCTTCCCAATAAGCATGTCTCC
*m-Cebpa*	CAAGAACAGCAACGAGTACCG	GTCACTGGTCAACTCCAGCAC
*m-Chrebp*	CCAGCCTCAAGGTGAGCAAA	CATGTCCCGCATCTGGTCA
*m-Cpt1a*	CTCCGCCTGAGCCATGAAG	CACCAGTGATGATGCCATTCT
*m-Cpt1b*	GCACACCAGGCAGTAGCTTT	CAGGAGTTGATTCCAGACAGGTA
*m-Cyp2e1*	CGTTGCCTTGCTTGTCTGGA	AAGAAAGGAATTGGGAAAGGTCC
*m-Fas*	TATCAAGGAGGCCCATTTTGC	TGTTTCCACTTCTAAACCATGCT
*m-Gapdh*	AGGTCGGTGTGAACGGATTTG	TGTAGACCATGTAGTTGAGGTCA
*m-Hsl*	CTGAGATTGAGGTGCTGTCG	CAAGGGAGGTGAGAGGGTAAC
*m-Lpin1*	CATGCTTCGGAAAGTCCTTCA	GGTTATTCTTTGGCGTCAACCT
*m-Ppara*	AGAGCCCCATCTGTCCTCTC	ACTGGTAGTCTGCAAAACCAAA
*m-Pparg*	TTTTCAAGGGTGCCAGTTTC	TTATTCATCAGGGAGGCCAG
*m-Scd1*	TTCTTGCGATACACTCTGGTGC	CGGGATTGAATGTTCTTGTCGT
*m-Srebf1*	GCAGCCACCATCTAGCCTG	GCAGCCACCATCTAGCCTG
*h-APOB*	GCAGGCCGAAGCTGTTTTG	GCACACGTTTCAGCCACTG
*h-CEBPA*	TGTATACCCCTGGTGGGAGA	TCATAACTCCGGTCCCTCTG
*h-FABP4*	ACTGGGCCAGGAATTTGACG	CTCGTGGAAGTGACGCCTT
*h-GAPDH*	GGAGCGAGATCCCTCCAAAAT	GGCTGTTGTCATACTTCTCATGG
*h-PPARG*	TACTGTCGGTTTCAGAAATGCC	GTCAGCGGACTCTGGATTCAG
*h-SREBF1*	ACAGTGACTTCCCTGGCCTAT	ACAGTGACTTCCCTGGCCTAT

*Apob*: apolipoprotein B, *Atgl*: patatin-like phospholipase domain containing 2, *Cd36*: CD36 molecules, *Cebpa*: CCAAT enhancer binding protein alpha, *Chrebp*: MLX interacting protein like, *Cpt1a*: carnitine palmitoyltransferase 1A, *Cpt1b*: carnitine palmitoyltransferase 1B, *Cyp2e1*: cytochrome P450 family 2 subfamily E member 1, *Fas*: fas cell surface death receptor, *Gapdh*: glyceraldehyde-3-phosphate dehydrogenase, *Hsl*: hormone-sensitive lipase, *Ppara*: peroxisome proliferator activated receptor alpha, *Pparg:* peroxisome proliferator activated receptor gamma, *Scd1*: stearoyl-Coenzyme A desaturase 1, *Srebf1:* sterol regulatory element binding transcription factor 1.

## Data Availability

The data presented in this study are available in insert article or Appendix A here.

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
