# Peer review of "Raphani Semen (Raphanus sativus L.) Ameliorates Alcoholic Fatty Liver Disease by Regulating De Novo Lipogenesis"

_nutrients, 2021, doi:10.3390/nu13124448_

Round 1

Reviewer 2 Report

In this study authors examined the pharmacological effects of water extract of Raphani Semen on alcoholic fatty liver disease (AFLD) in vitro and in vivo in a mouse model of AFLD (NIAAA). Water extract of Raphani Semen decresed the mRNA expression of de novo lipogenesis-related genes as well the respective protein levels of these factors.

The quality of the article is fair. Data obtained aim to provide alternative and complementary medical therapies with the regulatory property of hepatic lipid metabolism to current pharmaceuticals for the treatment of AFLD.

MAJOR REVISION: The background of the study is well described even if results are not clearly and logically presented. The authors must clarify methods and results sections to avoid confusion. The paper would benefit from stylistic changes to the way it has been written for a stronger, clearer, and more compelling argument

Material and Methods

The research method is appropriate to the objectives of the study but Authors should improve the flow and readability of the section, to present these elements as clearly and as logically as possibly.

  1. Authors should briefly comment on the choice of this aqueous, non-alcoholic, extraction
  2. Please specify in this section, not in discussion, which reference standard was used for the establishment of the mouse model for AFLD

100.Authors should better describe the experimental design. It is not very clear how many days the animals take RSWE (10 days?).  Duration, dosage (100mg / kg) and the low animal number for each group (n=4) should be better described in the methods section not in the results. In my opinion, the experimental procedure represented in Figure 1A should also be revised accordingly

  1. Authors should report the percentage of CO2 used for mice euthanasia.The AVMA Panel on Euthanasia recommends the use of CO2 at 10% to 30% CRR. As long as animals do not experience pain (mice are unconscious before CO2 reaches the pain-inducing level of 40%), it is recommend using faster CO2 CRR to decrease the time mice are experiencing distress.

Results

Figure 1. The histological panel should be enlarged for better visibility

  1. Authors should avoid from commenting on the data obtained in the results section

Figure 3A/.   Although the data refer to the cytotoxicity of RSWE, the bar graph shows the percentage of cell viability

Discussion

  1. Since of the association between AFLD and FA increase, the authors did not consider evaluating the possible reduction of total fatty acids after taking RSWE?
  2. To evaluate liver function, ALT and AST parameters are insufficient. Authors should at least evaluate LDH

Author Response

Response to Reviewer 2 comments

Comments and Suggestions for Authors

In this study authors examined the pharmacological effects of water extract of Raphani Semen on alcoholic fatty liver disease (AFLD) in vitro and in vivo in a mouse model of AFLD (NIAAA). Water extract of Raphani Semen decresed the mRNA expression of de novo lipogenesis-related genes as well the respective protein levels of these factors.

The quality of the article is fair. Data obtained aim to provide alternative and complementary medical therapies with the regulatory property of hepatic lipid metabolism to current pharmaceuticals for the treatment of AFLD.

MAJOR REVISION: The background of the study is well described even if results are not clearly and logically presented. The authors must clarify methods and results sections to avoid confusion. The paper would benefit from stylistic changes to the way it has been written for a stronger, clearer, and more compelling argument

Material and Methods

The research method is appropriate to the objectives of the study but Authors should improve the flow and readability of the section, to present these elements as clearly and as logically as possibly.

Q1. Authors should briefly comment on the choice of this aqueous, non-alcoholic, extraction

Answer: Traditional use of RS is usually prepared by boiling with hot water. This is the main reason why we have chosen the aqueous extraction method over ethanol extraction. Additionally, studies (PMID: 25467201; PMID: 26721217) report the anti-inflammatory effect of the water extract of RS. we added the relevant information to the Materials and Methods section 2.2. Thank you.

Q2. Please specify in this section, not in discussion, which reference standard was used for the establishment of the mouse model for AFLD

Answer: Per your request, we decided to transfer this sentence in results section 2.4 in the revised manuscript. Thank you.

Q3. 100.Authors should better describe the experimental design. It is not very clear how many days the animals take RSWE (10 days?).  Duration, dosage (100mg / kg) and the low animal number for each group (n=4) should be better described in the methods section not in the results. In my opinion, the experimental procedure represented in Figure 1A should also be revised accordingly

Answer: Per your request, we decided to describe the experimental design clearly about the date of administration and the number of mice per group, and change to the newly described Figure 1A in the revised manuscript. Thank you.

Q4. Authors should report the percentage of CO2 used for mice euthanasia.The AVMA Panel on Euthanasia recommends the use of CO2 at 10% to 30% CRR. As long as animals do not experience pain (mice are unconscious before CO2 reaches the pain-inducing level of 40%), it is recommend using faster CO2 CRR to decrease the time mice are experiencing distress.

Answer: We used up to 30% concentration of CO2 to euthanasia, and we wrote this in the revised manuscript. Thank you.

Results

Q5. Figure 1. The histological panel should be enlarged for better visibility

Answer: We have corrected. Thank you.

Q6. Authors should avoid from commenting on the data obtained in the results section

Answer: Per your request, we decided to delete in revised manuscript. Thank you.

Q7. Figure 3A/. Although the data refer to the cytotoxicity of RSWE, the bar graph shows the percentage of cell viability

Answer: We have corrected. Thank you.

Discussion

Q8. Since of the association between AFLD and FA increase, the authors did not consider evaluating the possible reduction of total fatty acids after taking RSWE?

Answer: We agree with reviewer’s point and decided to measure hepatic FFA levels in liver tissues. As a result, hepatic FFA levels were increased to 86.43 ± 21.33 compared to LD-fed mice (54.73 ± 11.63, p=0.057, n = 4). Meanwhile, hepatic FFA levels of RSWE group were decreased to 62.75 ± 5.96 (n = 4). This result has attached the results to Figure 1F. Thank you.

Q9. To evaluate liver function, ALT and AST parameters are insufficient. Authors should at least evaluate LDH

Answer: Per your request, we decided to LDH assay in PA-induced steatosis HepG2 cells, and attached the results to Figure 3C. Thank you.

Round 2

Reviewer 2 Report

Authors answered all requests made in the first review. The additional experiments strength to the work and results underlying their discussion are sufficiently supportive.